**nature** COMMUNICATIONS

# Photo-thermo-induced room-temperature phosphorescence through solid-state molecular motion

Xing Wang Liu[1,5], Weijun Zhao[2,5], Yue Wu [1✉], Zhengong Meng[1], Zikai He [3], Xin Qi[1], Yiran Ren[1], Zhen-Qiang Yu [1✉] & Ben Zhong Tang [4✉]

The development of smart-responsive materials, in particular those with non-invasive, rapid responsive phosphorescence, is highly desirable but has rarely been described. Herein, we designed and prepared a series of molecular rotors containing a triazine core and three bromobiphenyl units: o-Br-TRZ, m-Br-TRZ, and p-Br-TRZ. The bromine and triazine moieties serve as room temperature phosphorescence-active units, and the bromobiphenyl units serve as rotors to drive intramolecular rotation. When irradiated with strong ultraviolet photoirradiation, intramolecular rotations of o-Br-TRZ, m-Br-TRZ, and p-Br-TRZ increase, successively resulting in a photothermal effect via molecular motions. Impressively, the photothermal temperature attained by p-Br-TRZ is as high as 102 °C, and synchronously triggers its phosphorescence due to the ordered molecular arrangement after molecular motion. The thermal effect is expected to be important for triggering efficient phosphorescence, and the photon input for providing a precise and non-invasive stimulus. Such sequential photo-thermo-phosphorescence conversion is anticipated to unlock a new stimulus-responsive phosphorescence material without chemicals invasion.

[1] College of Chemistry and Environmental Engineering, Shenzhen University, Shenzhen 518071, China. [2] Feringa Nobel Prize Scientist Joint Research Center, School of Chemistry and Molecular Engineering, East China University of Science and Technology, Shanghai 200237, China. [3] School of Science, Harbin Institute of Technology, HIT Campus of University Town, Shenzhen 518055, China. [4] School of Science and Engineering, Shenzhen Institute of Aggregate Science and Technology, The Chinese University of Hong Kong, Shenzhen 518172, China. [5] These authors contributed equally: Xing Wang Liu, Weijun Zhao. ✉email: wuyue@szu.edu.cn; zqyu@szu.edu.cn; tangbenz@cuhk.edu.cn

Room-temperature phosphorescence (RTP) is luminescence that originates from the radiative transition from excited triplet state to ground state. Traditional inorganic RTP materials, which typically require noble or rare earth metals, have some intrinsic problems, including high cost, potential toxicity, and instability in aqueous environments[1–3]. It is thus essential to develop environmentally friendly, metal-free pure organic RTP materials[4,5]. Quantum yield and lifetime are the two critical indices for evaluating the performance of RTP materials. Thanks to the enthusiasm of many scientists, pure organic RTP systems with strong emission and long afterglow have been developed using many different strategies, including crystal engineering[6–9], host-guest interactions[10], H-aggregation[11–13], and polymer-matrix assistance[14]. Smart-responsive RTP materials, in which the RTP responds to external stimuli, have been described less often but are attracting increasing attention because of their widespread applications in sensing[15], imaging[16–18], and anticounterfeiting[19].

Because the excited triplet state is extremely susceptible to external stimuli, purely organic RTP is an excellent platform for constructing stimulus-responsive luminescent materials[20–23], which can be controllably tuned by external stimuli, including acid[24], mechanical force[25,26], thermal annealing[27], and magnetic or electric fields[28]. For the most part, thermal annealing is an effective way of enhancing RTP emission[29–32] but none of these strategies is ideal for regulating RTP properties, since the accumulation of chemicals or difficulty in carrying out the thermal annealing process obviously restricts potential applications. There is thus a significant opportunity to circumvent this bottleneck in the development of rapid and non-invasive stimulus-responsive RTP materials[33–35] by providing a straightforward and general route to smart-responsive materials[36–38].

Light is the optimal external trigger since photon input is not only clean and waste-free but also offers precise spatiotemporal control[39–43]. Herein, we describe a judicious photothermal strategy for the preparation of smart-responsive RTP materials, which uses photon input as a precise and non-invasive stimulus and takes advantage of the thermal effect to trigger efficient RTP. We have designed and prepared a series of molecular rotors containing a triazine core and three bromobiphenyl units: o-Br-TRZ, m-Br-TRZ, and p-Br-TRZ (Fig. 1a). The triazine moiety serves as a phosphorescence-active unit[44], and the bromine-bromobiphenyl units serve as rotors to drive intramolecular rotation. Following irradiation with intense UV light, intramolecular rotations of o-Br-TRZ, m-Br-TRZ, and p-Br-TRZ increase, resulting in photothermal effects via nonradiative decay. Interestingly, this type of intramolecular rotation between two benzene rings can be tuned by altering the position of the bromine substituent on the outer benzene ring. Successively elevated photothermal temperatures, produced via nonradiative decay, were seen on going from o-Br-TRZ to m-Br-TRZ to p-Br-TRZ. The photothermal effect could be harnessed to create smart photo-thermo-induced RTP materials (Fig. 1b), with the following extraordinary features: (i) ultrahigh photothermal temperatures (up to 102 °C), (ii) precise spatiotemporal RTP regulation (OFF-ON) via photothermal treatment, and (iii) high-level information processing. As far as we know, this type of report, that RTP can be triggered by a photothermal effect caused by molecular motion, has not been reported previously. The sequential photo-thermo-RTP conversion takes advantage of waste-free photo input and efficient thermo-induced molecular rearrangement.

## Results

**Photo-activated room-temperature phosphorescence.** The photophysical properties of triazine derivatives p-Br-TRZ, m-Br-TRZ, and o-Br-TRZ were firstly explored through a photo trigger.

As shown in Fig. 2a–c, o-Br-TRZ emits cyan luminescence, whereas m-Br-TRZ and p-Br-TRZ emit intense blue fluorescence before UV irradiation. When excited by weak 365 nm UV light in the solid state, the emission of o-Br-TRZ comprises two peaks: one at 394 nm, representing fluorescence, and one at 517 nm, representing RTP, whereas m-Br-TRZ and p-Br-TRZ emit only at 420 nm (Fig. 2d–f). In the solid state, all of the compounds emit intense fluorescence, with lifetimes in the range of 0.2–1.5 ns (Supplementary Fig. 1), indicative of room-temperature fluorescence. The lifetime of o-Br-TRZ emission at 517 nm is 117.1 μs at 298 K and increases to 218.4 ms at 77 K (Supplementary Fig. 2). These observations suggest that the emission peak at 517 nm is attributable to phosphorescence, with an RTP quantum yield of 1.47%, rather than to thermally activated delayed fluorescence. In addition to this difference in emission, o-Br-TRZ also shows very distinctive absorption behaviors upon UV excitation at room temperature. As shown in Fig. 2g–i, m-Br-TRZ and p-Br-TRZ, which absorb strongly in the UV region (<400 nm), are colorless. In contrast, o-Br-TRZ, which has another absorption band at 400–500 nm, is yellow in daylight, indicating that o-Br-TRZ has a highly ordered molecular arrangement and close molecular packing than m-Br-TRZ and p-Br-TRZ and is in favor of producing RTP.

We used powder X-ray diffraction (XRD) to investigate the mechanism of RTP by o-Br-TRZ. The XRD patterns of m-Br-TRZ and p-Br-TRZ powders show almost no visible diffraction peaks, indicating that they have amorphous characteristics, caused by mechanical grinding (Supplementary Fig. 3). In contrast, the XRD pattern of o-Br-TRZ powder shows sharp diffraction peaks, even after mechanical grinding, indicating the highly ordered molecular arrangement of o-Br-TRZ. Taken together with its yellow color, which results from the extensive conjugation of o-Br-TRZ, these results show that the solid-state luminescence of triazine derivatives is strongly governed by their ordered molecular arrangement, and that the RTP property of o-Br-TRZ could be attributed to a restrained nonradiative relaxation through close molecular packing[45–48].

Since photoirradiation is known to alter intermolecular interactions and cause dynamic changes in phosphorescence through molecular motion[49–53], we investigated the changes in the phosphorescence of o-Br-TRZ, m-Br-TRZ, and p-Br-TRZ powders after photoirradiation. The three powders were irradiated with UV light (365 nm, 516 mW/cm²) for 90 s to obtain irradiated o-Br-TRZ, m-Br-TRZ, and p-Br-TRZ. Essentially no change in the emission color of o-Br-TRZ was observed after irradiation (Fig. 2a). Compared with the untreated powders, irradiated m-Br-TRZ and p-Br-TRZ produced luminescence of different colors, which could easily be distinguished by the naked eye (Fig. 2b, c). The photoluminescence (PL) spectrum of o-Br-TRZ was nearly unchanged after UV irradiation (Fig. 2d), whereas the emission spectra of m-Br-TRZ and p-Br-TRZ showed significant changes (Fig. 2e, f). In addition to the initial emission peak at around 420 nm, irradiated m-Br-TRZ produced a new RTP emission peak at 511 nm, resulting in a change in luminescence color from blue to green. The phosphorescence quantum yield also increased from nearly 0% to 2.01% (Fig. 2b). Irradiated p-Br-TRZ showed an RTP emission peak at around 542 nm, leading to a change in luminescence color from blue to yellow, with a phosphorescence quantum yield of 2.94% (Fig. 2c). The lifetimes of irradiated o-Br-TRZ, m-Br-TRZ, and p-Br-TRZ were prolonged from 117.1, 3.3, and 1.3 μs to 131, 9.7, and 11.8 μs, respectively, representing 1.1-, 2.9-, and 9.1-fold increases, respectively (Supplementary Fig. 4). Furthermore, we performed theoretical calculations to investigate the intersystem crossing of o-Br-TRZ, m-Br-TRZ, and p-Br-TRZ, which prove that these three compounds all can emit RTP from the

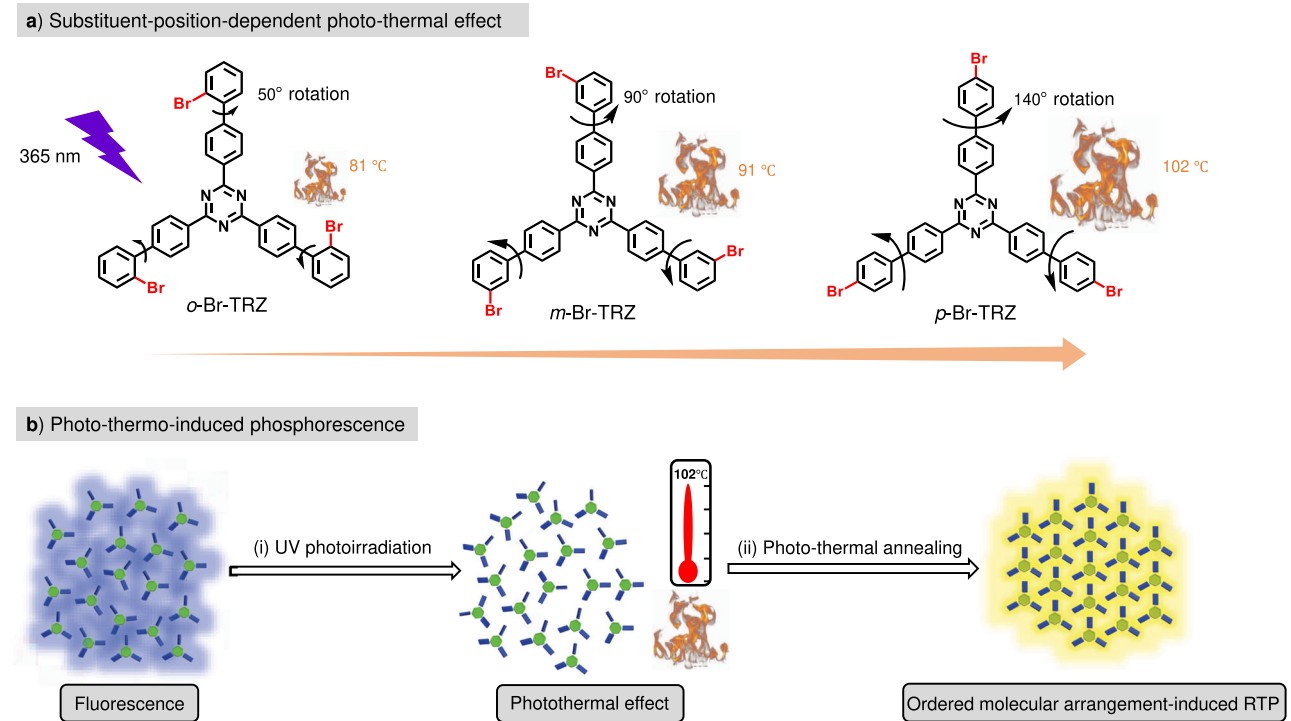

**Fig. 1 Molecular structures and photo-thermo-induced phosphorescence. a** Rational design of molecular rotors and successively elevated photothermal temperatures on going from *o*-Br-TRZ to *m*-Br-TRZ to *p*-Br-TRZ. **b** Sequential photo-thermo-induced phosphorescence (taking *p*-Br-TRZ as the example): (i) when irradiated with 365 nm UV light, *p*-Br-TRZ shows a photothermal effect through molecular motion, with a peak temperature of 102 °C; (ii) schematic representation of ordered molecular arrangement-induced phosphorescence through photothermal annealing, accompanied by ordered molecular arrangement.

perspective of energy gap and spin-orbit coupling (SOC) between the singlet and triplet state (Supplementary Fig. 5 and Supplementary Table 1). Noticeably, the RTP of irradiated *m*-Br-TRZ and *p*-Br-TRZ possesses decent stability, exhibits very tiny recessions, and nearly does not thermally convert back to the initial state (Supplementary Figs. 6 and 7). Similar to emission spectra, the reflection spectrum of irradiated *o*-Br-TRZ was nearly unchanged (Fig. 2g). Meanwhile, the reflection spectra of *m*-Br-TRZ and *p*-Br-TRZ both showed a new peak at ~435 nm (Fig. 2h, i), indicating the enhanced molecular packing than before UV irradiation. These super increases demonstrate that, especially for *m*-Br-TRZ and *p*-Br-TRZ, photoirradiation is an effective method for triggering RTP and prolonging lifetime, thus enabling rapid and non-invasive smart-responsive RTP materials. We carefully check the high-performance liquid chromatography spectra of *o*-Br-TRZ, *m*-Br-TRZ, and *p*-Br-TRZ powders before and after photoirradiation for 90 s (Supplementary Figs. 8–10). There are not any new peaks found in the spectra after photoirradiation. Besides, the photo-induced RTP behavior endows good reversibility, and the phototriggering and grinding processes could be repeated for several cycles without obvious fatigue (Supplementary Fig. 11).

**Photo-thermo-induced phosphorescence.** To probe the mechanism underlying this unique RTP "OFF-ON" process, we carried out control experiments, theoretical calculations, and analyses of XRD and emission spectra. Energy decay after excitation can be classified as nonradiative decay (heat, molecular motions) and radiative decay (luminescence). Significant efforts have been devoted to studying luminescence behavior (radiative decay), whereas less attention has been paid to the photothermal effects (nonradiative decay) in RTP triggered by photoirradiation. There is, therefore, ample scope for further investigation of the decay process from the excited state to the ground state.

In addition to their optical properties, we also investigated the photothermal properties of the triazine derivatives and, gratifyingly, uncovered a remarkable photothermal effect. Upon irradiation at 365 nm (516 mW/cm²), the temperature (*T*) of *o*-Br-TRZ responded dynamically to irradiation time, reaching a peak at 81 °C after approximately 15 s (Fig. 3a). When the light source was switched off, the temperature of the sample returned to room temperature after 15 s. In line with their RTP performance, the peak temperature of the photothermal effect increased sequentially from 81 °C for *o*-Br-TRZ to 91 °C for *m*-Br-TRZ and 102 °C for *p*-Br-TRZ. When excited by 365 nm light, the partial excited-state energies of all three isomers are thus lost nonradiatively in the form of heat via intramolecular rotations, leading to the observed photothermal effects. We concluded that the differences in photothermal temperatures can be attributed to the position of the bromine substituent on the outer benzene ring. Besides, only weak photothermal effects and nearly no RTP for *m*-Br-TRZ and *p*-Br-TRZ were observed by longer-wavelength photoirradiation such as 470 and 520 nm (Supplementary Figs. 12 and 13).

After assigning potentials to each atom, we carried out molecular dynamics simulations to investigate the mechanism of the photothermal gap for *o*-, *m*-, and *p*-Br-TRZs. Atomic coordinates and torsion angle statistics between two benzene rings were calculated. As shown in (Fig. 3b), there are big differences in molecular rotation between the three isomers. In the powdered state, the rotation angle of *o*-Br-TRZ lies mainly between 0° and 50° (probability: 81%), resulting in approximately 50° of rotation amplitude ($A \approx 50°$), the rotation angle of *m*-Br-TRZ ranges from −45° to 45°, resulting in a torsion amplitude of approximately 90° ($A \approx 90°$), and *p*-Br-TRZ molecules have the biggest rotation amplitude, ranging from −70° to 70° ($A \approx 140°$). These successive increases in torsion amplitudes ($A_{o\text{-Br-TRZ}} < A_{m\text{-Br-TRZ}} < A_{p\text{-Br-TRZ}}$)

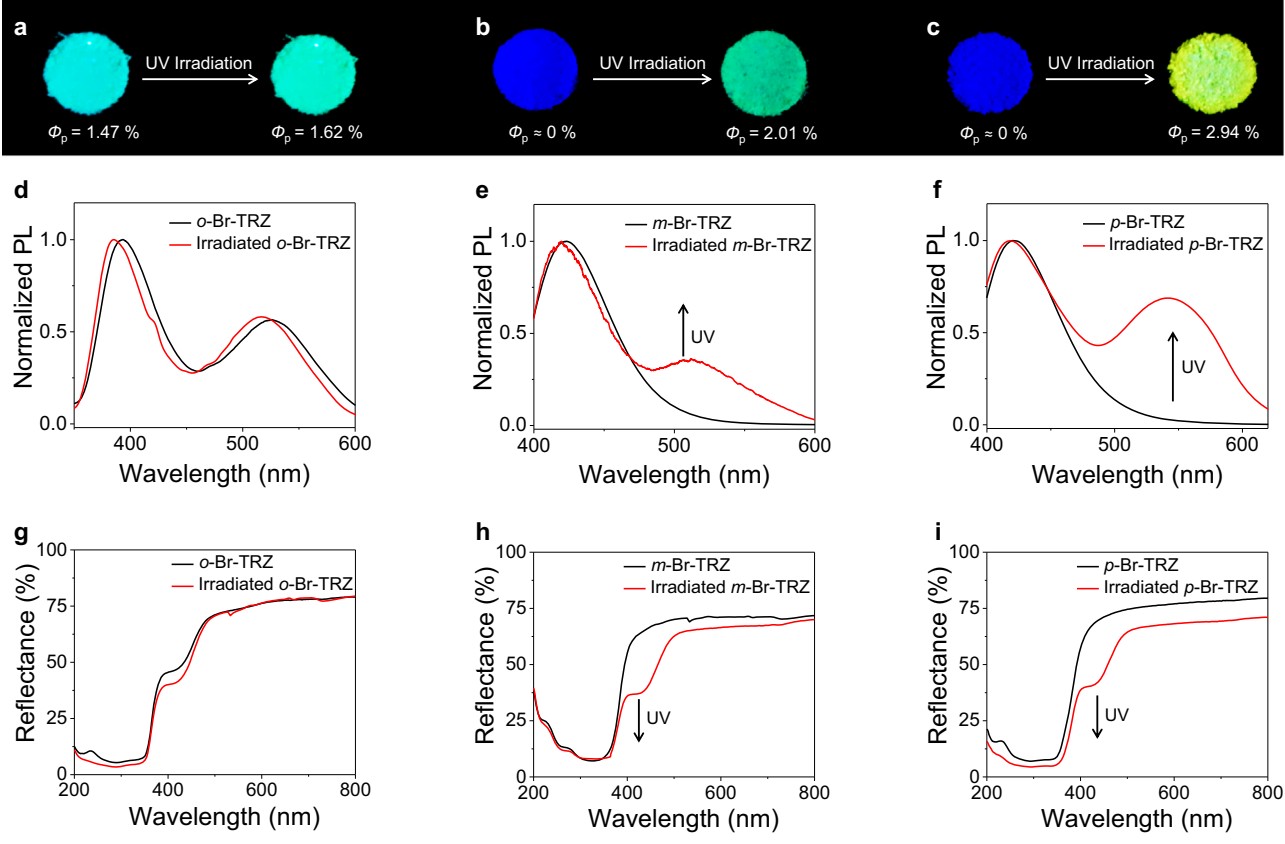

**Fig. 2 Photophysical properties of designed triazine derivatives.** Photos showing photoluminescence of (**a**) o-Br-TRZ, (**b**) m-Br-TRZ, and (**c**) p-Br-TRZ, before and after UV irradiation (365 nm LED, 516 mW/cm$^2$) for 90 s, with the change of phosphorescence quantum yield ($\Phi_p$). Photoluminescence spectra of (**d**) o-Br-TRZ, (**e**) m-Br-TRZ, and (**f**) p-Br-TRZ, before and after UV irradiation (365 nm LED, 516 mW/cm$^2$) for 90 s. In addition to the intrinsic RTP of o-Br-TRZ, RTP of m-Br-TRZ and p-Br-TRZ can be triggered by UV irradiation. UV-Vis diffuse reflectance spectra of (**g**) o-Br-TRZ, (**h**) m-Br-TRZ, and (**i**) p-Br-TRZ, before and after photoactivation under ambient conditions.

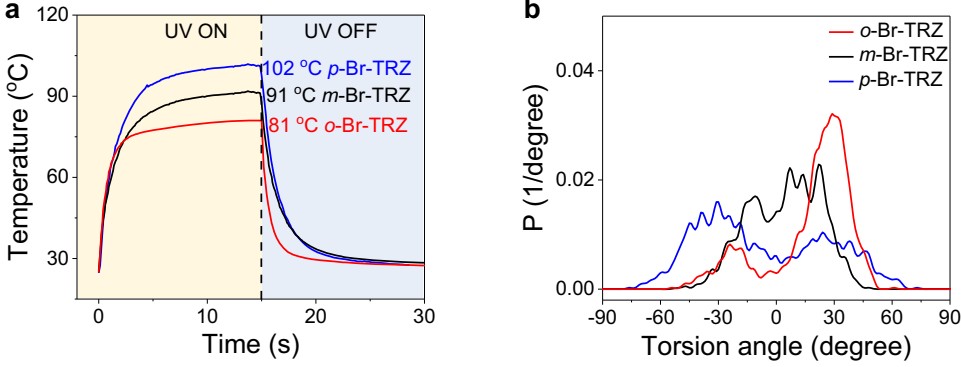

**Fig. 3 Photothermal effects of triazine derivatives. a** Bromine position-dependent photothermal temperature changes indicate that the photothermal effect is strongly correlated with substituent (Br) position and that p-Br-TRZ is best for photothermal conversion. Note: with irradiation at 365 nm (516 mW/cm$^2$), peak temperatures of the sample (5 mg, diameter of a round cake = 5 mm) were reached within 15 s. **b** Statistical probability of dynamic torsion angles between two benzene rings for p-Br-TRZ, m-Br-TRZ, and o-Br-TRZ.

thus show a positive correlation with the photothermal temperatures ($T_{o\text{-}Br\text{-}TRZ} < T_{m\text{-}Br\text{-}TRZ} < T_{p\text{-}Br\text{-}TRZ}$).

To further investigate the photo-thermo-induced phosphorescence, p-Br-TRZ was thermally annealed at different temperatures without UV irradiation and its phosphorescence spectra were recorded after the sample had cooled to room temperature (Fig. 4a). The thermogravity analysis results show that the 2% decomposition temperature of all the samples is higher than 440 °C and all the samples are highly stable during the thermal

treatment (Supplementary Fig. 14). On increasing the annealing temperature from 25 to 150 °C, the maximum phosphorescence wavelength showed a red shift from 525 to 562 nm, and the phosphorescent intensity of p-Br-TRZ at 562 nm displayed a marked, steady enhancement, suggesting a rapid response in phosphorescence to the elevated temperature of thermal annealing. The temperature-dependent ratio between phosphorescence intensity at 562 nm and fluorescence intensity at 422 nm increased sharply from 0.02 to 2.63, representing a 132-fold

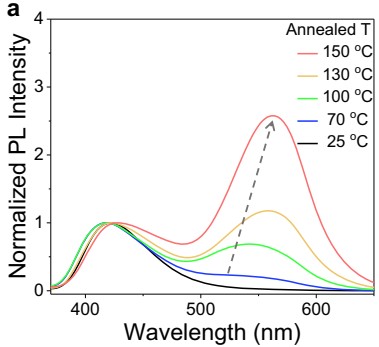
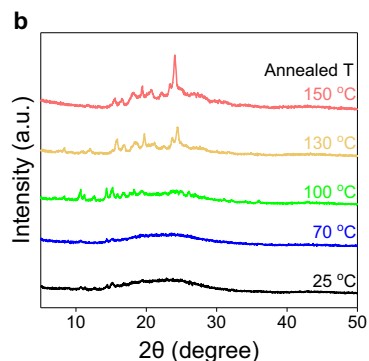
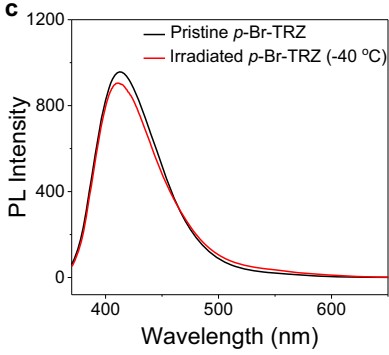

**Fig. 4 RTP activated by thermal effect. a** Emission and (**b**) XRD spectra of p-Br-TRZ (solid powder) recorded at 25 °C upon different heating treatments for 5 min at 25 (black line, no heating), 70 (blue line), 100 (green line), 130 (orange line), and 150 °C (red line), respectively. **c** Emission spectra obtained before and after UV irradiation (365 nm LED, 516 mW/cm$^2$) for 90 s at −40 °C.

increase. The XRD patterns suggest that the enhancement in phosphorescence after the thermal annealing process is attributable to the ordered molecular arrangement via molecular motion. The multiple peaks in the wide-angle range, especially the increases in the intensity of the diffraction peak at $2\theta = 23°$ in the XRD spectrum suggest that the ordered molecular arrangement of p-Br-TRZ enhances when the annealing temperature increases from 25 to 150 °C (Fig. 4b). Due to a cold crystallization peak at 77 °C, the exothermic peak in differential scanning calorimetry (DSC) curve of p-Br-TRZ during heating also indicates that there exists an ordered intermolecular rearrangement ranging from 77 to 151 °C (Supplementary Fig. 15). These results indicate that the phosphorescence stems from the more ordered molecular arrangement after thermal annealing[54]. m-Br-TRZ showed similar RTP, XRD, and DSC results to p-Br-TRZ, whereas no changes were observed in the emission or XRD spectra of o-Br-TRZ, and the DSC result also shows that there is no intermolecular rearrangement process during heating (Supplementary Figs. 15 and 16).

We next studied the luminescence of UV-irradiated p-Br-TRZ at low temperature to exclude heating effects. When we irradiated p-Br-TRZ powder at −40 °C with 365 nm light (516 mW/cm$^2$) for 90 s, almost no phosphorescence was detected (Fig. 4c) and the luminescence spectrum was unchanged from that of pristine p-Br-TRZ. These results demonstrate that UV irradiation cannot directly activate the phosphorescence of p-Br-TRZ, and that the "OFF-ON" phosphorescence is highly dependent upon the ordered molecular arrangement brought about by the photothermal effect, which can be regarded as a key index for photon-activated RTP in this study.

**Information processing based on sequential photo-thermo-phosphorescent conversion.** Having established the characteristics of the photo-thermo-induced phosphorescence process, we turned our attention to the effect of changes in the power of the irradiating light. We speculated that nonradiative decay by p-Br-TRZ may be enhanced by increasing 365 nm light LED power, which should increase molecular arrangement order and thus phosphorescence. When the power of 365 nm irradiation was increased, the photothermal temperature increased from 25 °C (no irradiation) to 101 °C (516 mW/cm$^2$, 90 s). Significantly, using Förster–Hoffmann equation fitting, we found that the peak temperature has a linear relationship with irradiation power ($R^2 = 0.997$, Fig. 5a), showing that p-Br-TRZ is capable of quantitatively sensing irradiation power.

As expected, the phosphorescence intensity of p-Br-TRZ (solid powder) at 542 nm showed a marked, steady, excitation power-

dependent enhancement, with an approximately 54-fold higher phosphorescence/fluorescence intensity ratio with 516 mW/cm$^2$ irradiation than with no irradiation, leading to a successive blue→yellow emission shift (Fig. 5b). This interesting phenomenon enables us to switch emission colors simply by changing the irradiation power, without the addition of any chemicals. The Commission Internationale de l'Eclairage (CIE) 1931 coordinates of p-Br-TRZ increased proportionally from (0.16, 0.06) to (0.24, 0.28) with increasing irradiation power (Fig. 5c), and the phosphorescence quantum yield increased from nearly 0% to 2.94%. As shown in (Supplementary Figs. 17 and 18), m-Br-TRZ also has the capacity for "OFF-ON" phosphorescence and a switch in luminescence color from blue to cyan. o-Br-TRZ, on the other hand, showed almost no variation in phosphorescence when the irradiation power was increased.

The unique dynamic phosphorescence properties of o-Br-TRZ, m-Br-TRZ, and p-Br-TRZ mean that they can be used in many interesting optical devices and applications, such as information encryption[55]. Because the three isomers show strong emission as solid powders, we explored the possibility of using them to generate an anticounterfeiting pattern on black paper using imprint lithography (Fig. 5d). All letters were created using blue dye naphthalene, except for "R" (p-Br-TRZ), "T" (m-Br-TRZ), and "P" (o-Br-TRZ). As illustrated in (Fig. 5d), the black paper showed a bright cyan "P" and other blue letters under 365 nm light. When irradiated with stronger 365 nm light (216 mW/cm$^2$), the letter "R" also appeared in cyan. Under moderate power light, two letters, "R" and "P", were thus distinguished from the other blue letters. When the irradiation power was increased to 516 mW/cm$^2$, the luminescence color of the letter "R" switched from cyan to yellow, and the previously blue letter "T" emitted cyan. The true information, yellow "R", cyan "T" and "P", could then be observed. These different characteristic outputs enable multilevel data encryption and decryption that is distinct from conventional approaches.

## Discussion

In summary, we have developed a series of bromine-substituted molecular rotors in which the color of photo-thermo-induced phosphorescence can be manipulated by varying the position of the bromine substituent and the irradiation power. Initially, because of the steric effect of the halogen substitution, o-Br-TRZ is highly ordered arrangement and produces bright phosphorescence by a molecular packing-induced phosphorescence mechanism, whereas the other isomers are amorphous and do not emit phosphorescence. When irradiated by strong UV light, intramolecular rotations of o-Br-TRZ, m-Br-TRZ, and p-Br-TRZ

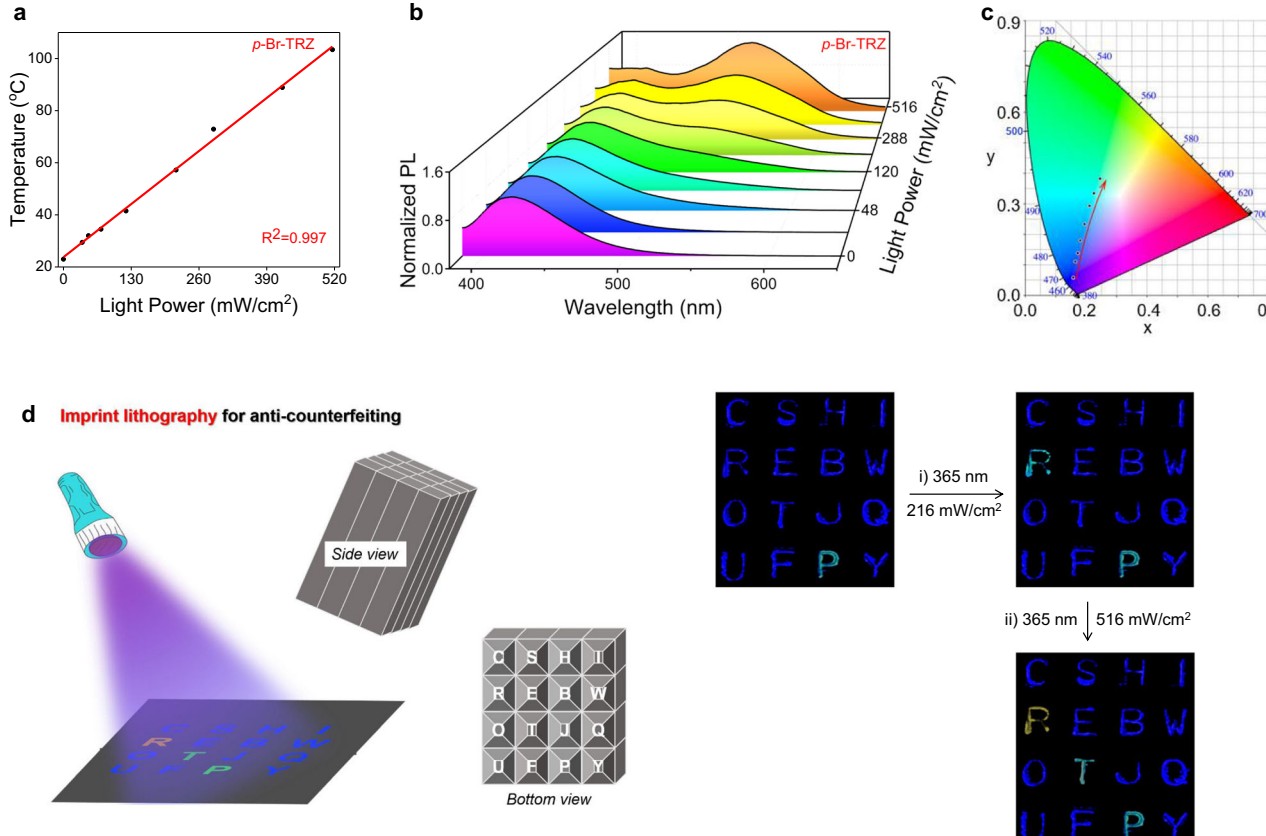

**Fig. 5 Light-power-response RTP and application. a** Changes in the photothermal temperature of powdered *p*-Br-TRZ on increasing the power of 365 nm light from 0 to 516 mW/cm$^2$ demonstrate a linear relationship between photothermal temperature and irradiation powder. **b** Three-dimensional plot of luminescence of *p*-Br-TRZ vs power of 365 nm light vs wavelength showing dynamics of enhancement of phosphorescence and phosphorescence/fluorescence intensity ratio on increasing irradiation power. **c** Corresponding 1931 CIE coordinate diagram of *p*-Br-TRZ with different power 365 nm irradiation, in accordance with (**b**). **d** Imprint lithography with luminescent dyes for a multilevel anticounterfeiting pattern. By modulating the irradiation light power (0→216→516 mW/cm$^2$), different luminescence colors can be recorded; all letters were prepared using commercially available blue dye naphthalene except for "R" (*p*-Br-TRZ), "T" (*m*-Br-TRZ), and "P" (*o*-Br-TRZ).

are increased, resulting in a photothermal effect via nonradiative decay. Their intrinsic sensitivity to molecular motions enables ordered molecular arrangements of *p*-Br-TRZ and *m*-Br-TRZ and allows "OFF-ON" phosphorescence. This sequential photo-thermo-phosphorescent conversion for non-invasive and smart-responsive RTP materials thus emerges, using the fundamental properties of molecular radiative and nonradiative decay.

## Methods

**Materials**. Chemicals were purchased from Energy-Chemical and used without further purification. Solvents and other common reagents were obtained from Energy-Chemical. Solvents were dried and distilled out before being used for the synthesis. $^1$H NMR and $^{13}$C NMR spectra were measured on Bruker ADVANCE III 600 MHz, Bruker ADVANCE III 500 MHz, Bruker ADVANCE III 400 MHz, and Varian INOVA 400 MHz spectrometers. High-resolution mass spectra were recorded on a Thermo Finnigan TSQ 7000 mass spectrometer. Elementary analysis was performed on Elementar vario EL cube. The starting materials and end products were synthesized following the procedures described in the Supplementary information. Powder XRD patterns were performed on a PANalytical Empyrean diffractometer with Cu Kα radiation at 25 °C (scan range: 5–50°). Gel filtration chromatography was performed using a ZORBAX SB-C18 column (Agilent) conjugated to an Agilent 1260 Infinity II Prime HPLC system. The flow rate was fixed at 1.0 mL/min, the injection volume was 20 μL and each sample was run for 10 min. The absorption wavelength used was set at 296 and 318 nm. Fourier transform infrared spectra were collected by using a Shimadzu IR Affinity-1 spectrometer and spectra were obtained from 4000 to 400 cm$^{-1}$. DSC measurements were carried out on a NETZSCH DSC-200F3 instrument at a heating rate of 10 °C min$^{-1}$ in nitrogen. TGA analysis spectra were investigated on Mettler Toledo TGA\DSC 3+. The detailed experimental procedure and synthetic methods are described in the Supplementary information.

**Steady-state spectral measurements**. The steady-state absorption measurements were recorded on a SHIMADZU UV-3600PLUS spectrophotometer. PL spectra were measured on Hitachi F 7000 and absolute luminescence quantum yields were measured on HAMAMATSU Quantaurus-QY C 11347-11 spectrofluorometer with an integrating sphere.

**Transient spectral measurements**. The fluorescence lifetime was measured using Edinburg FLS-980 and phosphorescence lifetime was measured using Edinburg FLS-1000 with a continuous xenon lamp (Xe2), a microsecond flash-lamp (uF1), a picosecond pulsed diode laser (EPLED-320), a detector (Single Photon Counting PMT), and an integrating sphere (N-M01-Integrating Sphere), respectively.

**Theoretical and computational methods**. All-electron density functional theory (DFT) calculations have been carried out by the latest version of ORCA quantum chemistry software (Version 5.0.1). For geometry optimization calculations of S$_1$ structure, wB97X-D3 functional and the def2-TZVP basis set were used, and the optimal S$_1$ geometry for each compound was determined. The DFT-D3 dispersion correction with BJ-damping was applied to correct the weak interaction to improve the calculation accuracy. The excited states and SOC calculations were performed with PBE0 functional and the DKH-def2-TZVP basis set. SOC calculation was performed by the spin-orbit mean-field method. To evaluate the torsion angle statistics between two benzene rings, long-range electrostatic interactions were accounted for using the Ewald method and the parameters for each like-site interaction were obtained using the COMPASS-II force field.

## Data availability

All the data supporting the findings in this work are available within the manuscript and Supplementary information file. Source data are available for Figs. 2–5 in the associated source data file. Source data are provided with this paper.

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

## Acknowledgements

B.Z.T. gratefully acknowledges support from NSFC/China (21700102), Z.-Q.Y. thanks the support from NSFC/China (21875143) and Innovation Research Foundation of

Shenzhen (JCYJ20180507182229597), Y.W. thanks the support from NSFC/China (21908146), W.Z. thanks the support from Shanghai Pujiang Program (20PJ1402900), Shanghai Science and Technology Commission Basic Project-Shanghai Natural Science Foundation (21ZR1418400). Z.H. thanks the support from Innovation Research Foundation of Shenzhen (GXWD20201230155427003-20200728150952003). Special thanks to the Instrumental Analysis Center of Shenzhen University (Lihu Campus).

## Author contributions

Y.W., Z.-Q.Y., and B.Z.T. conceived the project and designed the experiments. X.W.L., W.Z., Y.W., X.Q., Y.R., and Z.-Q.Y. were primarily responsible for the data collection and analysis. X.W.L., W.Z., Y.W., Z.M., and Z.H. analyzed the RTP data. X.W.L., W.Z., and Y.W. prepared the figures and wrote the original manuscript text. Y.W., Z.-Q.Y., and B.Z.T. completed the manuscript. All the authors contributed to the discussions and manuscript preparation.

## Competing interests

The authors declare no competing interests.
