## [Peer Review File · Nature Communications]

Photo-thermo-induced room-temperature phosphorescence through solid-state molecular motionReviewers' Comments:

Reviewer #1:

Remarks to the Author:

In this work, Liu and co-workers reported a series of photo-induced RTP materials based on triazine derivatives. Experimental results, especially for the thermal annealing process, indicate that the photo-thermo-induced molecular motion should be mainly responsible for it. This phenomenon is interesting, but more works are still needed to fully disclose it. Thus, I recommend it to be published after revision.

1. In Figure 4, the authors attributed the thermal-annealing/photo-induced RTP emission to the ordered molecular arrangement. Then, how about the RTP effect of p-Br-TRZ and m-Br-TRZ crystals?
2. In this work, the authors attributed the RTP effect of these target compounds to the crystallization-induced phosphorescence mechanism, which is too general. It is believed that the particular molecular packing has played significant role in RTP effect, which could be partially inferred from the changed UV-vis after UV-irradiation in Fig 2. The authors should make more discussions about it.
3. If this photo-induced RTP effect could be reversible under other external stimuli?
4. The thermal properties for the three target compounds should be studied, including DSC and TGA.
5. In lines 158-161, the rotation angles and the corresponding rotation amplitudes for the three target compounds were presented. What was the selection criteria? According to Fig. 3b, the rotation angle of o-Br-TRZ seems to be from -50 to 50°. The authors should make a more detail presentation about it.
6. Could other irradiation light with longer wavelength lead to the photo-induced RTP effect in this work?

Reviewer #2:

Remarks to the Author:

The paper "Photo-thermo-induced room-temperature phosphorescence through solid-state molecular motion" by B. Z. Tang et al. reports new smart responsive materials based on room temperature phosphorescence triggered by UV-light irradiation. The idea behind this new study is brilliant, the research is nicely put within the context of the existing literature and the experimental work is thoroughly carried out. This is an excellent paper and the results add significantly to this field of research. The main findings seem to fulfil the urgency and novelty requirements of Nature Communications. However, before accepting this manuscript for publication, the following minor points should be considered.

- 1) I wonder whether the behavior observed for the meta and para compounds (photo-thermo-induced RTP) is then exhibited indefinitely by the investigated materials or if it is reversible after some time and can be eventually activated again after further irradiation. Did the authors check the optical properties of the obtained materials over time?
- 2) I wonder whether the authors have any information about the InterSystem Crossing efficiency for the three molecules by for instance transient absorption measurements. I would be curious to know whether the only difference is in the phosphorescence or even the triplet production is different for the ortho relative to the meta/para compounds.
- 3) Page 5, lines 117-118 and relative Supplementary Figure 4. If I understand well, the authors did observe long phosphorescence decays for m-Br-TRZ and p-Br-TRZ also before UV irradiation. If so, the only effect of photo-irradiation is the lengthening of the emission lifetimes and enhancement of the emission intensity? In Figure 2 the authors report the phosphorescence quantum yield to be 0% before UV-irradiation for these samples. I am confused...
- 4) The authors found from the XRD measurements a crystalline nature of the o-Br-TRZ powder, which may be the reason underlying the observed RTP of this material even before UV-irradiation. Can the authors go deeper into this peculiar behavior? Is it possible to suggest something as the reason for this in connection with the different molecular structure?

5) The Methods section is quite poor. For instance, can the authors give more details about the light sources used for the time resolved fluorescence measurements? Or about the molecular dynamic simulations?

6) The authors should carefully check their reference list. For instance: n. 22 and 53 refer to the same paper....

Response to the Reviewers' comments

Reviewer 1

In this work, Liu and co-workers reported a series of photo-induced RTP materials based on triazine derivatives. Experimental results, especially for the thermal annealing process, indicate that the photo-thermo-induced molecular motion should be mainly responsible for it. This phenomenon is interesting, but more works are still needed to fully disclose it. Thus, I recommend it to be published after revision.

Reply: We sincerely thank the reviewer for the appreciation and recognition concluding that this article deserves publication in *Nature Communications*. We also greatly appreciate the reviewer for the very careful review, and our manuscript has been improved according to the reviewer's suggestion.

1. In Figure 4, the authors attributed the thermal-annealing/photo-induced RTP emission to the ordered molecular arrangement. Then, how about the RTP effect of *p*-Br-TRZ and *m*-Br-TRZ crystals?

Reply: Thanks for the valuable comment of the reviewer.

While irradiated by 365 nm light irradiation, new absorption bands ranging from 400–500 nm emerged (Figure 2h-i), indicating that the enhanced intermolecular packing of *m*-Br-TRZ and *p*-Br-TRZ through photothermal annealing. Therefore, the nonradiative decay was restrained due to the enhanced molecular packing, resulting in the increased RTP of *m*-Br-TRZ and *p*-Br-TRZ. Meanwhile, the XRD patterns of *p*-Br-TRZ (Fig. 4b) and *m*-Br-TRZ (Supplementary Fig. 16d) also exhibit strong signals after thermal annealing, further verifying the ordered molecular arrangement. We tried to obtain the crystal of *p*-Br-TRZ and *m*-Br-TRZ but failed, which could be attributed to three heavy bromine atoms.

Besides, we tried mechanical grinding to destroy the ordered molecular arrangement of irradiated *p*-Br-TRZ, accompanying with an intensively eliminated RTP (Supplementary Fig 11a). It further verified that the enhanced molecular packing could be a key factor for triggering RTP.

Supplementary Figure 11a. PL spectra of irradiated *p*-Br-TRZ before and after mechanical grinding.

Furthermore, we performed theoretical calculations to investigate their InterSystem Crossing (ISC) on RTP, whose methods and details are shown in Experimental Methods and Supplementary Information. **Supplementary Fig. 5** shows that *o*-Br-TRZ, *m*-Br-TRZ and *p*-Br-TRZ all have three possible channels for ISC based on the same transition orbital compositions between T_n and S_1 within the ± 0.3 eV energy level (Fraser et al. J. Am. Chem. Soc. 2007, 129, 8942; Bunz et al. J. Am. Chem. Soc. 2013, 135, 2160–2163). It is very coincident that all the spin-orbit coupling (SOC) between S_1 and T_3 energy level ($\xi_{S_1T_3}$) for *o*-Br-TRZ, *m*-Br-TRZ and *p*-Br-TRZ have the highest value with small energy gaps (ΔE_{ST}), becoming the most efficient ISC channel (**Supplementary Fig. 5 and Table 1**). The $\xi_{S_1T_3}$ of *o*-Br-TRZ, *m*-Br-TRZ and *p*-Br-TRZ is 2.956, 2.311, and 2.073 cm^{-1} , respectively. These theoretical calculations prove that they all can emit RTP from the perspective of ΔE_{ST} and SOC, and the enhanced molecular packing (restriction of their nonradiative relaxation) could be a critical index for emitting RTP.

Supplementary Figure 5. Calculated energy diagram of S_1 and T_n , spin-orbit coupling for the involved S_1 and T_n states of molecules.

Supplementary Table 1. The energy level and spin-orbit coupling of *o*-Br-TRZ, *m*-Br-TRZ, and *p*-Br-TRZ.

	S_1-T_n	ΔE_{ST} (eV)	ξ (cm^{-1})	$\xi / \text{exp}(\Delta E_{ST}^2)$
o-Br-TRZ	S_1-T_2	0.214	0.022	0.021
	S_1-T_3	0.067	2.956	2.944
	S_1-T_4	0.121	1.514	1.499
m-Br-TRZ	S_1-T_2	0.212	0.052	0.051
	S_1-T_3	0.064	2.311	2.302
	S_1-T_4	0.012	1.169	1.169
p-Br-TRZ	S_1-T_2	0.212	0.296	0.285
	S_1-T_3	0.103	2.073	2.052
	S_1-T_4	0.022	1.234	1.234

2. In this work, the authors attributed the RTP effect of these target compounds to the crystallization-induced phosphorescence mechanism, which is too general. It is believed that the particular molecular packing has played significant role in RTP effect, which could be inferred from the changed UV-vis after UV-irradiation in Fig 2. The authors should make more discussions about it.

Reply: We thank the reviewer for the extremely professional comments to improve our manuscript. As mentioned in Question 1, we think that the enhanced molecular packing (restricted nonradiative relaxation) is a key factor for RTP behavior in our system. In

consideration of absorption change upon UV irradiation (Fig 2e-f), RTP elimination after mechanical grinding (Supplementary Fig. 11a), and theoretical calculation on ISC (Supplementary Fig. 5 and Table 1), we totally agree with the reviewer that the enhanced molecular packing has played significant role in RTP effect. (Pages 3-5, 10 in Manuscript)

3. If this photo-induced RTP effect could be reversible under other external stimuli?

Reply: Thanks very much for the suggestion of the reviewer.

According to the reviewer's suggestion, we found that mechanical force could effectively eliminate RTP. As shown in Supplementary Fig 11, the I_{542}/I_{417} value of irradiated *p*-Br-TRZ was 0.69. After mechanical grinding for 60 s, the RTP dismissed intensively and I_{542}/I_{417} ratio decreased to 0.08, resulting in a weakened RTP of *p*-Br-TRZ. Meanwhile, the photo-induced RTP behavior endows good reversibility, and the phototriggering and grinding processes could be repeated for several cycles without obvious fatigue (Page 6 in Manuscript).

Supplementary Figure 11. (a) PL spectra of irradiated *p*-Br-TRZ before and after mechanical grinding. (b) Photo-induced RTP of *p*-Br-TRZ endows good fatigue resistance between photoirradiation and mechanical grinding.

4. The thermal properties for the three target compounds should be studied, including DSC and TGA.

Reply: Thanks for the suggestion. We conducted the TGA and DSC experiments in N_2 and the results are shown in the following. The 2% decomposition temperature of *o*-Br-TRZ and *p*-Br-TRZ is higher than 440 °C, while that of *m*-Br-TRZ is even higher than 480 °C, indicating all

the three compounds have good thermal stability (Page 7 in Manuscript and Supplementary Fig. 14). During heating, the exothermic transition at 58 and 77 °C, and endothermic transition at 146 and 151 °C for *m*-Br-TRZ and *p*-Br-TRZ are observed, respectively, which mean that there exist intermolecular rearrangement processes ranging from 58 to 146 °C (*m*-Br-TRZ), and from 77 to 151 °C (*p*-Br-TRZ). For *o*-Br-TRZ, we only observed an endothermic transition at about 90 °C during heating (Page 7 in Manuscript and Supplementary Fig. 15).

Supplementary Figure 14. TGA curves of the three compounds in N₂ at a heating rate at 20 °C/min.

Supplementary Figure 15. DSC curves of (a) *o*-Br-TRZ, (b) *m*-Br-TRZ, and (c) *p*-Br-TRZ during heating at a heating rate of 10 °C/min.

5. In lines 158-161, the rotation angles and the corresponding rotation amplitudes for the three target compounds were presented. What was the selection criteria? According to Fig. 3b, the rotation angle of *o*-Br-TRZ seems to be from -50 to 50°. The authors should make a

more detail presentation about it.

Reply: We greatly appreciate the reviewer for his/her careful review.

Though the rotation angle of *o*-Br-TRZ ranges from -50° to 50° , the rotation probability of *o*-Br-TRZ below 0° is quite small, whereas probability of rotation angle lying between 0° to 50° is calculated as nearly 81%. Thus, the mainly torsion of *o*-Br-TRZ mainly ranges from 0° to 50° . (Page 6 in Manuscript)

6. Could other irradiation light with longer wavelength lead to the photo-induced RTP effect in this work?

Reply: According to the reviewer's suggestion, we conducted 470 and 520 nm light irradiation for *m*-Br-TRZ and *p*-Br-TRZ. Compared with 365 nm light irradiation, 470 nm light irradiation-induced photothermal temperature were 41 and 44 °C corresponding to *m*-Br-TRZ and *p*-Br-TRZ, respectively, whereas irradiated by 520 nm, the photothermal temperature was even lower (Supplementary Fig. 12). In the condition of lower photothermal temperatures, nearly no RTP was observed in luminescence spectra for *m*-Br-TRZ and *p*-Br-TRZ (Supplementary Fig. 13). Thus it is difficult for longer wavelength lights to lead to the photo-induced RTP effect in our work (Page 6 in Manuscript).

Supplementary Figure 12. Photothermal effect of (a) *m*-Br-TRZ and (b) *p*-Br-TRZ upon 470 (515 mW/cm²) and 520 nm (512 mW/cm²) irradiation.

Supplementary Figure 13. PL spectra of (a) *m*-Br-TRZ and (b) *p*-Br-TRZ upon 470 (515 mW/cm²) and 520 nm (512 mW/cm²) irradiation for 90 s.

Reviewer 2

The paper “Photo-thermo-induced room-temperature phosphorescence through solid-state molecular motion” by B. Z. Tang et al. reports new smart responsive materials based on room temperature phosphorescence triggered by UV-light irradiation. The idea behind this new study is brilliant, the research is nicely put within the context of the existing literature and the experimental work is thoroughly carried out. This is an excellent paper and the results add significantly to this field of research. The main findings seem to fulfil the urgency and novelty requirements of Nature Communications. However, before accepting this manuscript for publication, the following minor points should be considered.

Reply: We greatly appreciate the reviewer for his/her positive comments concluding that this article deserves publication in *Nature Communications*, especially for “The idea behind this new study is brilliant, the research is nicely put within the context of the existing literature and the experimental work is thoroughly carried out. This is an excellent paper and the results add significantly to this field of research”.

We also greatly appreciate the reviewer for the very careful review, and our manuscript has been improved according to the reviewer’s suggestion.

1. I wonder whether the behavior observed for the meta and paracompounds (Photo-induced RTP) is then exhibited indefinitely by the investigated materials or if it is reversible after some time and can be eventually activated again after further irradiation. Did the authors check the optical properties of the obtained materials over time?

Reply: Thanks for the valuable comment from the review. According to the reviewer’s suggestion, we conducted a series of time-dependent RTP tests for checking the stability of RTP of *m*-Br-TRZ and *p*-Br-TRZ.

The RTP of irradiated *m*-Br-TRZ and *p*-Br-TRZ both possesses decent thermal stability, and exhibits very tiny recessions and nearly do not thermally convert back to the initial state as demonstrated by the constant I_{511}/I_{420} (*m*-Br-TRZ) and I_{542}/I_{417} (*p*-Br-TRZ) ratio during 10

days (Page 5 in Manuscript and Supplementary Fig. 6 and 7). To answer the reviewer’s query “if it is reversible after some time and can be eventually activated again after further irradiation”, we used mechanical grinding to accelerate the RTP recession of irradiated *p*-Br-TRZ, and the 365 nm light irradiation could recover its RTP property again (Supplementary Fig. 11a). Repeatedly toggling the RTP on and off using 365 nm irradiation and grinding showed that photo-induced RTP has good fatigue resistance (Supplementary Fig. 11b).

Supplementary Figure 6. (a) PL spectra of irradiation *m*-Br-TRZ before and after 10 days. (b) Luminescence intensity ratio (I_{511}/I_{420}), nearly no reversal change was observed after 10 days, indicating that irradiated *m*-Br-TRZ is stable.

Supplementary Figure 7. (a) PL spectra of irradiation *p*-Br-TRZ before and after 10 days. (b) Luminescence intensity ratio (I_{542}/I_{417}), nearly no reversal change was observed after 10 days, indicating that irradiated *p*-Br-TRZ is stable.

Supplementary Figure 11. (a) PL spectra of irradiated *p*-Br-TRZ before and after mechanical grinding. (b) Photo-induced RTP of *p*-Br-TRZ endows good fatigue resistance between photoirradiation and mechanical grinding.

2. I wonder whether the authors have any information about the InterSystem Crossing efficiency for the three molecules by for instance transient absorption measurements. I would be curious to know whether the only difference is in the phosphorescence or even the triplet production is different for the ortho relative to the meta/para compounds.

Reply: Thanks very much for the reviewer's professional and kind suggestion.

According to the reviewer's suggestion, we tried ultra-fast transient absorption spectroscopy to investigate the InterSystem Crossing (ISC) efficiency of *o*-Br-TRZ, *m*-Br-TRZ, and *p*-Br-TRZ. Due to the messy scattering light for solid powders and molecular damages by using strong lasers, we failed to obtain useful data. For better addressing the reviewer's question. We performed theoretical calculations on these three compounds, whose methods and details are shown in Experimental methods and Supplementary information. According to the perturbation theory, the rate constant (k_{ISC}) of ISC is given by:

$$k_{ISC} \propto \langle {}^1\Psi | \hat{H}_{SO} | {}^3\Psi \rangle / \exp(\Delta E_{ST}^2) \quad (1)$$

where $\langle {}^1\Psi | \hat{H}_{SO} | {}^3\Psi \rangle$ is the spin-orbit coupling (SOC, ξ) matrix element, and ΔE_{ST} is the energy gap between the singlet and triplet states. This equation suggests that large SOC and small ΔE_{ST} can result in high k_{ISC} . **Supplementary Fig. 5** shows that *o*-Br-TRZ, *m*-Br-TRZ and *p*-Br-TRZ all have three possible channels for ISC based on the same transition orbital compositions between T_n and S_1 within the ± 0.3 eV energy level (Fraser et al. J. Am. Chem.

Soc. 2007, 129, 8942; Bunz et al. J. Am. Chem. Soc. 2013, 135, 2160). It is very coincident that all the $\xi_{S_1T_3}$ of *o*-Br-TRZ, *m*-Br-TRZ and *p*-Br-TRZ have the highest value with small ΔE_{ST} , becoming the most efficient ISC channel (Page 5 in Manuscript and Supplementary Fig. 5 and Table 1). The $\xi_{S_1T_3}/\exp(\Delta E^2_{S_1T_3})$ of *o*-Br-TRZ, *m*-Br-TRZ and *p*-Br-TRZ is 2.944, 2.302, and 2.052, respectively, with small gaps. According to the equation 1 on k_{ISC} , the ISC rate constant is $k_{ISC(o-Br-TRZ)} > k_{ISC(m-Br-TRZ)} > k_{ISC(p-Br-TRZ)}$ with small difference.

Supplementary Figure 5. Calculated energy diagram of S1 and Tn, spin-orbit coupling for the involved S1 and Tn states of molecules.

Supplementary Table 1. The energy level and spin-orbit coupling of *o*-Br-TRZ, *m*-Br-TRZ, and *p*-Br-TRZ.

	S_1-T_n	ΔE_{ST} (eV)	ξ (cm ⁻¹)	$\xi / \exp(\Delta E^2_{ST})$
o -Br-TRZ	S_1-T_2	0.214	0.022	0.021
	S_1-T_3	0.067	2.956	2.944
	S_1-T_4	0.121	1.514	1.499
m -Br-TRZ	S_1-T_2	0.212	0.052	0.051
	S_1-T_3	0.064	2.311	2.302
	S_1-T_4	0.012	1.169	1.169
p -Br-TRZ	S_1-T_2	0.212	0.296	0.285
	S_1-T_3	0.103	2.073	2.052
	S_1-T_4	0.022	1.234	1.234

3. Page 5, lines 117-118 and relative Supplementary Figure 4. If I understand well, the authors did observe long phosphorescence decays for *m*-Br-TRZ and *p*-Br-TRZ also before UV irradiation. If so, the only effect of photo-irradiation is the lengthening of the emission lifetimes and enhancement of the emission intensity? In Figure 2 the authors report the phosphorescence quantum yield to be 0% before UV-irradiation for these samples. I am confused....

Reply: Thanks for the reviewer's careful revision and sorry for our unclear description.

Actually, we cannot observed RTP for initial *m*-Br-TRZ and *p*-Br-TRZ in Figure 2b-c and 2e-f, indicating that their RTP quantum yields are closely approaching 0% (Pages 5 and 9). Therefore, photoirradiation could trigger RTP on a macroscale. Indeed, their lifetime decays were calculated as 3.3 and 1.3 μ s, thus we use " \approx " rather than "=" symbols in Fig. 2b-c.

4. The authors found from the XRD measurements a crystalline nature of the *o*-Br-TRZ powder, which may be the reason underlying the observed RTP of this material even before UV-irradiation. Can the authors go deeper into this peculiar behavior? Is it possible to suggest something as the reason for this in connection with the different molecular structure?

Reply: Thanks very much for your professional and kind suggestion.

While irradiated by 365 nm light irradiation, new absorption bands ranging from 400–500 nm emerged (Figure 2h-i), indicating that the enhanced intermolecular packing of *m*-Br-TRZ and *p*-Br-TRZ through photothermal annealing. Therefore, the nonradiative decay was restrained due to the close molecular packing, resulting in the increased RTP of *m*-Br-TRZ and *p*-Br-TRZ. Beyond the absorption spectra, the XRD patterns of *m*-Br-TRZ (Supplementary Fig. 16d) and *p*-Br-TRZ (Fig. 4b) also exhibit obvious signals after thermal annealing, further verifying the more ordered molecular arrangement. Furthermore, the XRD pattern of initial *o*-Br-TRZ displays sharp diffraction peaks ranging from 7 to 37 degree (2θ) without photoirradiation or thermal annealing, indicating the highly ordered molecular arrangement. Combining with an absorption band at 400–500 (Fig. 2g), it demonstrates that the

restrained nonradiative relaxation is in connection with the RTP property of *o*-Br-TRZ (Page 5).

5. The Methods section is quite poor. For instance, can the authors give more details about the light sources used for the time resolved fluorescence measurements? Or about the molecular dynamic simulations?

Reply: Thanks for the constructive comment from the reviewer. According to his/her kind suggestion, we added some experimental details on XRD, HPLC, FTIR, DSC, TGA, light source, and theoretical calculation in methods section (Page 11).

6. The authors should carefully check their reference list. For instance: n. 22 and 53 refer to the same paper....

Reply: We greatly appreciate the reviewer for his/her careful review. We rechecked the reference and cited four new papers:

Ref. 8: Mieno, H., Kabe, R., Notsuka, N., Allendorf, M. D. & Adachi, C. Long-lived room-temperature phosphorescence of coronene in zeolitic imidazolate framework ZIF-8. *Adv. Opt. Mater.* **4**, 1015–1021 (2016).

Ref. 13: Zhou, B. & Yan, D. Hydrogen-bonded two-component ionic crystals showing enhanced long-lived room-temperature phosphorescence via TADF-assisted Förster resonance energy transfer. *Adv. Funct. Mater.* **29**, 1807599 (2019).

Ref. 23: Hirata, S., Totani, K., Yamashita, T., Adachi, C. & Vacha, M. Large reverse saturable absorption under weak continuous incoherent light. *Nat. Mater.* **13**, 938–946 (2014)

Ref. 53: Wagner, P. J., May, M. J., Haug, A. & Graber, D. R. Phosphorescence of phenyl alkyl ketones. *J. Am. Chem. Soc.* **92**, 5269–5270 (1970).

Reviewers' Comments:

Reviewer #1:

Remarks to the Author:

The article has been greatly improved after revision and I suggest it to be published at current form.

Reviewer #2:

Remarks to the Author:

The authors have addressed my comments and requests and I believe that the paper can now be published.